| Editor's Pick | Microbial Ecology | Perspective

# Advancing microbiome research in Māori populations: insights from recent literature exploring the gut microbiomes of underrepresented and Indigenous peoples

Ella T. Silk,[1,2] Simone B. Bayer,[1,2] Meika Foster,[2,3] Nicole C. Roy,[2,4,5] Michael W. Taylor,[2,6] Tommi Vatanen,[7,8,9,10] Richard B. Gearry[1,2]

**ABSTRACT** The gut microbiome plays vital roles in human health, including mediating metabolism, immunity, and the gut-brain axis. Many ethnicities remain underrepresented in gut microbiome research, with significant variation between Indigenous and non-Indigenous peoples due to dietary, socioeconomic, health, and urbanization differences. Although research regarding the microbiomes of Indigenous peoples is increasing, Māori microbiome literature is lacking despite widespread inequities that Māori populations face. These inequities likely contribute to gut microbiome differences that exacerbate negative health outcomes. Characterizing the gut microbiomes of underrepresented populations is necessary to inform efforts to address health inequities. However, for microbiome research to be culturally responsible and meaningful, study design must improve to better protect the rights and interests of Indigenous peoples. Here, we discuss barriers to Indigenous participation in research and the role disparities may play in shaping the gut microbiomes of Indigenous peoples, with a particular focus on implications for Māori and areas for improvement.

**KEYWORDS** human microbiome, gut microbiome, Indigenous health, social disparities, diet, urbanization, Maori health, Indigenous research

Microorganisms that reside within human microbiomes are essential for host physiology and health (1). Human microbiomes are composed of bacteria, archaea, eukaryotes, and viruses. Bacteria are particularly abundant, inhabiting the human body at a roughly 1:1 ratio to human cells (2). The gut, with $10^{14}$ bacterial cells, harbors the vast majority of commensals; it is the most species-rich microbiome of the human body and displays considerable variation among individuals, highlighting the environmental, context, and host-specific nature of this microbiome (3).

Gut microbiome research has advanced rapidly in the last two decades due to increased recognition of the importance of gut microbes to human health, desire to understand the implications of the significant variation in microbiome composition, and, above all, remarkable technical advances in high-throughput sequencing coupled with reducing costs (4). The human gut microbiome is involved in regulating dietary nutrient digestion, host metabolism, and the immune system, including protection against infection by pathogens (5–7). Due to its involvement in diverse physiological interactions, microbial dysbiosis is associated with a multitude of different physical and mental disorders, for example, cardiovascular disease and diabetes mellitus (plus their associated risk factors), cancer, gastrointestinal diseases or disorders (such as IBD and IBS) and infections, autism, anxiety, and depression (8–12). These associations emphasize the interrelationships between the gut microbiome, quality of human life, and well-being.

Despite intense recent scrutiny, many factors, likely to influence gut microbiome composition, still require further elucidation. One major example is ethnicity, with a

**Peer Reviewers** Aviâja Lyberth Hauptmann, Ilisimatusarfik, Nuuk, Greenland; Kieran O'Doherty, University of Guelph, Guelph, Canada

Address correspondence to Richard B. Gearry, richard.gearry@otago.ac.nz.

The authors declare no conflict of interest.

See the funding table on p. 18.

pressing need to extend microbiome research to a greater diversity of peoples. Some recent literature suggests that ethnicity has a significant effect on gut microbiota composition, even when different ethnicities live in the same geographical location (13). Despite this, there is relatively little research with Indigenous populations, including those residing in developed countries. People who identify as Indigenous make up approximately 6% (~476 million people) of Earth's population but comprise 19% of the extreme poor, suffering greater socioeconomic and educational disparities than non-Indigenous ethnicities (14–16). Although Indigenous populations are often the most at risk of health disparities, they are still chronically underrepresented in health research. Furthermore, the unique social contexts and rich cultural heritage of Indigenous peoples (e.g., languages, connections to the environment, and Indigenous knowledge systems) and the historical crises that many have faced (e.g., colonization, cultural genocide) are all likely to have influenced microbe-human relationships over time yet are not well considered in the literature thus far (17). Researchers should give careful thought to the selection of research questions that are most relevant and beneficial to Indigenous populations (17, 18).

Māori are the Indigenous people of Aotearoa New Zealand (NZ), with approximately 17% of the population identifying as Māori (19). Like many Indigenous peoples, Māori are disproportionately affected by disease, particularly non-communicable diseases, such as type 2 diabetes mellitus (T2DM) and cardiovascular disease (20, 21). Although there is evidence that such diseases are linked to the gut microbiome (22, 23), there are limited studies involving Indigenous populations and the gut microbiome, and currently no studies examining gut microbiomes of NZ Māori. Māori have a long history of scientific research and have recently been leading efforts to decolonize research in Aotearoa NZ, which is likely to lead to more successful and educational Indigenous health research (24–26). This is especially necessary in research that explores the connection between nature (i.e., gut microbes) and human health, as this topic aligns well with Indigenous world views, which encompass a holistic understanding of well-being and appreciation of unseen entities essential for good health (18).

The limited studies analyzing the gut microbiomes of Indigenous peoples are important as they establish differences not only between Indigenous and non-Indigenous populations but also within Indigenous populations (27–37). Microbial differences relate to numerous environmental, physical, and social factors such as diet (which is both a source of microbes and provides nourishment for existing gut microbes), socioeconomic status (SES), and health, all of which are further impacted by urbanization. These inter-connected factors influence each other while also impacting the gut microbiome and are likely to explain much of the variance between gut microbiomes of Indigenous and non-Indigenous people (Fig. 1). More gut microbiome research is urgently needed in Indigenous populations to better understand this important mediator of health and disease in different contexts. The studies available to date have shown that it is inappropriate to simply extrapolate and generalize data from non-Indigenous to Indigenous populations. This perspective piece discusses reasons why there is notably limited literature in this field, and how Māori ideologies could improve gut microbiome research. We further discuss literature of relevance to gut microbiomes of Indigenous peoples, urbanization, diet, SES, and health disparities. The primary author of this research (ES) is a descendant of the Iwi (tribes) Ngāti Kahu and Ngāpuhi, and a coauthor (MF) is a descendant of Te Ātiawa and Ngāti Mutunga; it is important to note that opinions reflected in this perspective are based on their experiences and world views and do not necessarily speak for all Māori and Indigenous peoples. Other co-authors (SB, NR, MT, TV, and RG) agree that working with Indigenous communities and targeting underrepresented groups is central to improving health outcomes and have expertise in the topics discussed in this perspective.

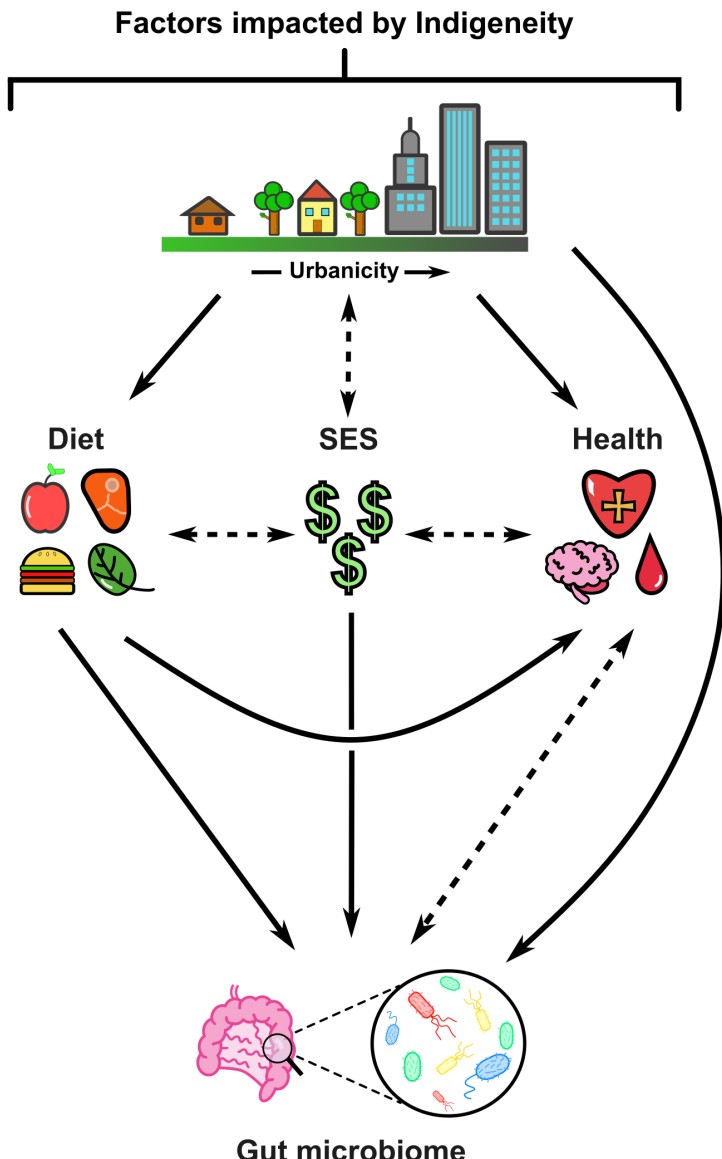

**Factors impacted by Indigeneity**

**Urbanicity**

**Diet**

**SES**

**Health**

**Gut microbiome**

**FIG 1** Diet, disparities, and urbanization shape the gut microbiomes of Indigenous peoples. Solid arrows represent hypothesized unidirectional link between factors. Dashed arrows represent hypothesized bidirectional links between factors. Figure generated using Inkscape.

## DEFINING INDIGENEITY IN RESEARCH

In research, Indigeneity has consistently been difficult to define. According to the United Nations, "Indigenous peoples and nations are those which, having a historical continuity with pre-invasion and pre-colonial societies that developed on their territories, consider themselves distinct from other sectors of the societies now prevailing on those territories, or parts of them." The United Nations further suggests that such populations usually harbor strong links to surrounding natural resources and territories and are, in the modern day, the non-dominant groups of society (38).

Although a formal definition exists for Indigenous peoples, and there is clearly variation between Indigenous and non-Indigenous peoples, Indigenous populations are themselves heterogeneous due to inter- and intra-population variability. Interpopulation variability between Indigenous ethnicities may be the primary driver of microbiome differences due to their spread across over 90 geographically distinct countries, leading to environmental and dietary differences and engagement in different cultural practices (16); however, intrapopulation variability combines social aspects, such as connection to culture, physical aspects like geographic location within country of residence (i.e., urban versus rural or remote living), and biological aspects, like genealogy (39). Many Indigenous people have genealogy that is mixed with other (often European) ethnicities, especially post-colonization. In many Indigenous cultures, how far back an individual's Indigenous heritage spans does not matter. For example, in Māori culture, the criterion for self-identification as Māori is simply having any Māori whakapapa (genealogy; relationships), and percentages of other ethnic genealogy do not make one more or less Māori (40).

On one hand, the self-identifiable nature of assigning indigeneity allows a more accurate picture of how many Indigenous people exist, as many Indigenous individuals do not harbor the physical traits stereotypically associated with indigeneity (41). In contrast, self-identification will not capture the many people with Indigenous genealogy who do not identify as Indigenous due to the lack of connection to their ancestry or culture or being unaware of their ancestry altogether (e.g., the stolen generations of Australia) (41). In the NZ context, it is reasonable to hypothesize that those who do not recognize or know about their 'Māori' heritage may suffer similar inequities to self-identified Māori due to previous colonization events that contribute to a cycle of poverty; in other words, due to effects that are class-based rather than outcomes of racism. Although the effects of class and racism often go hand-in-hand, there is debate as to which has the biggest impact on Māori in poverty (42), both of which change influential environmental factors like SES (further impacting living conditions), diet, and health, and therefore gut microbial ecology (Fig. 1). These considerations are not stated to divide members of Indigenous communities or to pass judgement on how people choose to self-identify but rather to point out potential causes in data variability among groups and recognize that there is not just one "Indigenous gut microbiome" profile.

## BOLSTERING RESEARCH ON GUT MICROBIOMES OF INDIGENOUS PEOPLES

In order to start to speculate about the gut microbiome profiles of Māori populations, this perspective piece will first discuss some of the limitations of current research and how non-Indigenous researchers can work more effectively with Indigenous peoples. Then, in the next section, we will specifically discuss how Māori concepts can be integrated into gut microbiome research in New Zealand, in an effort to improve health inequities. Finally, we will draw upon examples of gut microbiome studies in Indigenous populations to highlight (i) variation between Indigenous versus non-Indigenous populations and (ii) variations within Indigenous populations based on urbanization and disparities.

Many of the gut microbiome studies that have been conducted in Indigenous populations to date are limited by low participant numbers, the provision of primarily observational data, and performing research that is not relevant to Indigenous communities and will not provide beneficial outcomes to them. This is not unique to microbiome research: Fig. 2 depicts some of the barriers to Indigenous people participating in health and clinical research, which highlight the need for researchers to form trusted relationships with potential Indigenous participants, to be able to clearly describe the benefits of the research for the Indigenous person or group, and to take into account cultural considerations and world views when designing and implementing the research. In addition, many Indigenous populations live in extremely remote, non-urbanized environments, which introduces a variety of recruitment and participation challenges, as well as implications for the generalizability of the research. For

10.1128/msystems.00909-24 **4**

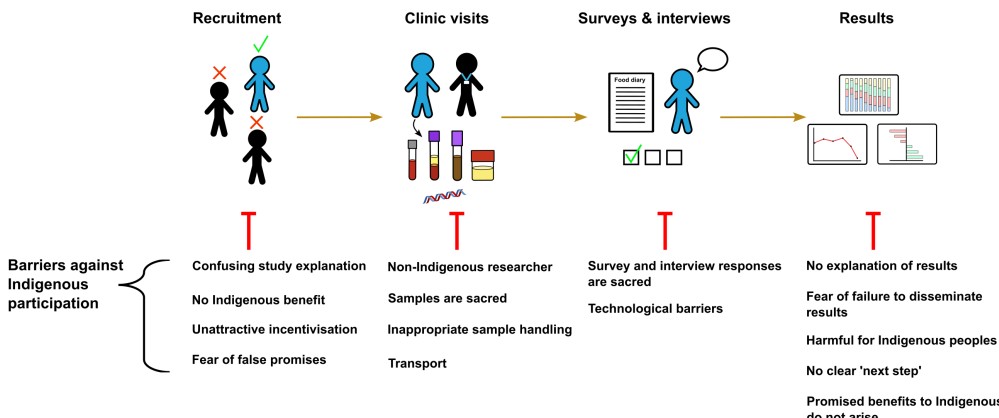

**FIG 2** Barriers against Indigenous participation in health research. Across the clinical trial process there are numerous physical and spiritual factors that hinder Indigenous participation. Targeting such barriers will allow more successful Indigenous research. Figure generated using Inkscape.

example, in many aspects, research in rural Indigenous populations may better reflect Indigenous realities prior to colonization; however, the findings of these studies are less likely to be generally applicable to westernized and urbanized Indigenous groups, necessitating dedicated research in these populations. It is also important to note that those who live in rural environments may not eat diets that align with those of their ancestors, and those who live in more urban environments might have better access to food that aligns with ancestral diets due to monetary advantages. This point highlights the importance of careful recruitment and characterization of research participants to ensure that findings are accurately described and reflect the research question.

"How do research institutions learn to better engage with Indigenous and other marginalized communities in microbiome research?" is a question of particular relevance to Māori (17). There is a clear need and potential for research in this space (and in Māori-based clinical research in general) due to the well-established inequities faced by Māori populations. In order to ensure maximum Māori participation in research of this nature and generate meaningful impact for communities, Māori microbiome research must be deemed relevant by Māori and designed to meet their own research aspirations (25, 43, 44). These criteria for engagement may also extend to other Indigenous populations due to many having similar values and beliefs to Māori. The specific Māori concepts discussed here can be found in Table 1.

Although education surrounding Māori worldviews and practices is becoming more prevalent in NZ, there remains major distrust between Māori people and non-Māori health professionals (45). Often, doctors and researchers do not meet the requirements of Māori in healthcare and research, especially regarding their ability to understand, incorporate, or align with tikanga Māori (customary system of values and practices) and spiritual dimensions in their approach. Furthermore, modern Māori health disparities originated largely due to colonization and can be explained by socioeconomic factors, westernization of the traditional Māori diet, and a multitude of introduced diseases (46, 47). These factors have led many Māori to adopt a preference for Māori-led research to be performed by researchers with Māori whakapapa. Known as Kaupapa Māori research, the main facet of this approach is that research is conducted "By Māori, for Māori" within NZ; it recognizes that Māori have the capability to design and conduct research that aligns with their values and prioritizes Māori interests and Mātauranga Māori (Māori knowledge systems) (25, 43, 44). The use of Kaupapa Māori and Mātauranga Māori approaches is likely to lead to improvements in recruitment and compliance of Māori participants in microbiome research, due to participants feeling more connected to the research

**TABLE 1** Discussed Māori terms, concepts, and their definitions

| Concept | Definition |
| --- | --- |
| Hauora | Health and wellness in the four pillars of health: physical, mental, social, and spiritual well-being. |
| Iwi | A Māori tribe associated with a distinct territory within New Zealand. |
| Kai | Food or a meal. |
| Kaupapa Māori | Taking a Māori approach to a topic or challenge that includes Māori views, Māori leadership, and aims to affirm Māori self-definitions by Māori for Māori. |
| Manaakitanga | An important value in Māori society, meaning to show kindness, respect, hospitality, and generosity toward guests. |
| Maramataka | The Māori lunar calendar, which marks the phases of the moon. Traditionally consulted for almost any activity taking place in a Māori community, including planting and harvesting food. The Maramataka also marks significant annual events and other seasonal activities. |
| Mātauranga Māori | The cultural knowledge system of Māori, which is grounded within the values, concepts, and practices that underpin Māori world views. |
| Mauri | A symbol, force, energy, and essence of life. This can be applied to people, flora and fauna, objects, and ecosystems. |
| Noa | Known, common, and free from restriction. A protocol that is integral to Māori life, impacting everyday activities; complementary to tapu. |
| Pūrākau | Māori narratives, storytelling, legends, communication, and transmission of ideas. |
| Rongoā | Traditional Māori medicine; a system of healing. |
| Tangata whenua | The people of the land. In the context of Aotearoa NZ, these are Māori. |
| Taonga | Treasure, anything prized, applied to anything considered to be of value including socially or culturally valuable objects, resources, phenomenon, ideas, and techniques. |
| Tapu | Unknown, sacred, prohibited, restricted, set apart. A protocol that is integral to Māori life, impacting everyday activities; complementary to noa. |
| Tikanga | Māori customary system of values and practices; correct procedures, protocols, and practices. |
| Whakapapa | Genealogy, relationships, and the connection between animate and inanimate things and the spiritual world. A principal concept in Māori culture. |
| Whanaungatanga | Family relationships, kinship, and a sense of connection. |

purpose and staff and due to the cultural acceptability of the research design. Where a Kaupapa Māori approach is impossible, meaningful consultation with Māori must be undertaken to determine which Māori populations are most likely to need and benefit from microbiome research and ideally how Mātauranga Māori can be incorporated to enhance the relevance of the research for Māori, rather than scientific curiosity being the main driver of the research direction and design.

## MĀTAURANGA MĀORI IN HUMAN GUT MICROBIOME RESEARCH

Mātauranga Māori has much to offer research, including that on microbiomes, and is often mistakenly viewed as non-factual, unquantifiable, and "not science" (48, 49). This debate limits the development of new knowledge, which is the purpose of science. Māori have a rich history of practicing science through observation, experimentation, and analysis of their environments, with extensive knowledge transmitted across generations related to, for example, land use, flora and fauna, biodiversity and ecological interrelationships, waterways, astrological bodies, weather patterns, and human and environmental well-being (24). Māori science is dynamic and has evolved in parallel to so-called "western" science (50), offering the potential for Mātauranga Māori research and its creative capacity to complement gut microbiome research (51).

Many Māori have a deep understanding of ecology and the importance of maintaining an ecological balance, particularly as it pertains to the Aotearoa NZ context due to their status as tangata whenua (people of the land) (24). Māori embrace a holistic view of how hauora (health and wellness) is connected to the wider ecosystem, which aligns well with the evolving scientific understanding of the roles of microorganisms of the body and the genes they contain. Specifically, gut microbes are implicated in nutrient absorption and the harvest, storage, and expenditure of energy obtained from the diet, allowing people to process and gain nutrition from foods harvested from the land and

highlighting the clear interrelationship between the ecosystem, natural resources, and the human body (18).

Whakapapa informs Indigenous knowledge, is central to Māori society, and can be used as a model to explain how all life forms are interconnected (51–53). In the same way that Māori recognize that human samples have whakapapa, microbes themselves hold whakapapa, having descent lines and demonstrating morphological and ecological relationships and functions, which coincides with modern day phylogeny and holobiont theories (multiple species existing in a symbiotic ecological unit) (52, 54). A related concept, whanaungatanga, emphasizes the associations within the natural world and is critical for understanding the interdependencies between humans and the environment, including the microbial environment (55). The application of Mātauranga Māori perspectives, such as the belief that all life moves as a functional unit, has the potential to enhance western microbial ecology theories (24), thereby creating new knowledge at the western science and Mātauranga Māori research interface (51).

It is important, however, to recognize and respect the boundaries and considerations that are inherent in research that involves Mātauranga Māori. In a Māori worldview, biological samples (tissue, DNA, RNA) derived from humans are taonga (treasures; prized) and tapu (sacred; unknown) and are imbued with whakapapa (53, 56–58). Different levels of tapu or restrictions may apply to different types of samples, with the head, blood, genitals, and buttocks area traditionally considered to carry more tapu (55, 59). A request to provide fecal samples for gut microbiome studies may therefore introduce significant cultural sensitivities for many Māori, perhaps more so than taking simple skin or mouth swabs for examination of the skin and oral microbiomes. Providing fecal samples has already presented itself as an issue for Māori and Pasifika non-responders for bowel cancer screening in NZ, with some mentioning tapu as a barrier (59). Oftentimes, participants are asked to store fecal samples in the fridge or freezer. As many Māori hold the view that fecal samples (tapu) should not be kept in a place where food (noa; known) is stored, this requirement would likely act as a major barrier against some Māori participating in gut microbiome studies (55, 60). For this reason, providing fecal sample kits that are stable at room temperature may be more appropriate for Indigenous participants, as well as ensuring that researchers understand the potential hesitance surrounding the gathering of these samples. Clinical studies should carefully cater to Māori populations, and consent should clearly address both the physical and spiritual relevancy of biological samples. The meaningful involvement of appropriately resourced Māori researchers and advisors will facilitate the application of tikanga practices during sample handling (e.g., karakia; incantations) to recognize and respect the tapu, mauri (life force), and whakapapa attached to human-derived samples (61). The successful incorporation of Kaupapa Māori practices into clinical trials and health research has been shown to lead to good engagement and positive health outcomes for Māori (42, 62–65), demonstrating the advantages of research being undertaken by research staff with experience in implementing culturally appropriate methodologies.

Data, photographs, and diary-like records (such as survey responses) are also considered to be taonga (56, 58). Many Māori are wary as to how Māori data from samples and responses are handled, stored, discarded, used, and published, which may limit their desire to participate in research (56). To decolonize the way in which Māori data are viewed, it is important to acknowledge that whakapapa and mauri remain present even in anonymized data (53, 56, 57). Furthermore, although anonymization may protect individual identities, it does not account for the significant risk of ethnic injustice that occurs when an Indigenous data set is made public, and any perceived negative information is generalized to the whole population. A prime example of this can be seen in the context of epigenetic research pertaining to monoamine oxidase A. Dubbed the "warrior" gene, the monoamine oxidase A gene, was found in higher proportions in Māori than in non-Māori NZ Europeans, leading to suggestions that there is a genetic predisposition for Māori to be aggressive and have addictive tendencies; this conceptual leap effectively minimizes any consideration of the ongoing effects of

environmental deprivation and historical trauma that Māori have been subjected to due to colonization (66, 67) and eroded Māori trust in research. Due to previous instances of ethnic injustice, working with and interpretation of Māori microbiological data require exceptional care and Māori leadership to avoid any harm and discrimination to Māori communities, particularly as open data and science are becoming the gold standard (68). An understanding of data sovereignty principles and implementation of good practice in Indigenous data governance should be mandatory for microbiome studies that involve Māori. The CARE (Collective benefit, Authority to control, Responsibility, Ethics) principles for Indigenous data governance provide current standards for ensuring that data are handled appropriately in research related to Indigenous populations (68, 69). These principles address privacy and future-use (e.g., biobanking) issues, which have been neglected previously (68, 69). Building and disseminating a repertoire of successful Māori research will provide a clearer picture of the importance of Māori data for Māori communities and may be used as a tool to inspire future Māori generations to contribute to microbiome research and continue to unravel existing research and health inequities and create solutions (25).

Although promoting Māori engagement in research is important to facilitate meaningful and constructive change, care must be taken to ensure that Māori scientists based within research institutions are not inadvertently or deliberately required to work a "cultural double shift," where it is assumed or expected that on top of their usual research load, they will take the lead in facilitating connections to Māori communities, even when they have previously had no experience making such connections (70). It is inappropriate to assume all Māori have connections to their whakapapa (or want involvement in Mātauranga Māori research). One example of a Māori-led institute in NZ that is successfully training young Māori to be self-determined empowered leaders who can bridge science and Mātauranga Māori knowledge systems is Pūhoro STEMM (Science, Technology, Engineering, Mathematics, Mātauranga) Academy. Pūhoro has developed a capability building pipeline that lifts Māori into the STEMM workforce by taking a whānau wrap-around approach to tutoring and learning and provides tertiary students with paid research internship opportunities (which often contain aspects of Mātauranga Māori) to expose young Māori to STEMM disciplines. The first author of this manuscript has been a Pūhoro STEMM Academy student for 8 years and is now a PhD candidate who has since co-supervised an undergraduate Pūhoro internship student, whereas a number of authors (NR, SB, MF, RG) have had roles supervising and advising Pūhoro students on their projects.

Dissemination of results in an understandable way is one of the most important ways of demonstrating the importance of research for hauora. Māori often find the participant information and results for studies to be complex and inaccessible (56). They often have an expectation of individualized results, which should be communicated appropriately for proper understanding and to show the importance of using their samples (56). Scientists tend to overcomplicate communication to lay populations, which may be particularly relevant for some Māori who hold different ideas of what constitutes science. Although science communication in Aotearoa is not new for Māori, who traditionally have sophisticated mechanisms and processes for the oral cross-generational transfer of knowledge (such as pūrākau), the development of communication in NZ parallels other English speaking countries where colonization has resulted in the diminishment of Indigenous language and ideologies (71). Prioritizing oral communication channels for the dissemination of results to Māori may allow for better understanding and the opportunity to get clarity by asking questions in person. Having more personalized and understandable dissemination may contribute to the formation of pūrākau science narratives, which can be disseminated further. Appropriate communication of microbial results is essential, as microbiome research is a very complex and rapidly developing field due to ongoing technological advancements.

As with other scientific fields, the recognition and incorporation of Indigenous knowledge systems and practices in gut microbiome research will benefit Indigenous

participants, communities, and researchers and is likely to lead to the formation of new knowledge. Making changes that enhance the engagement and trust of Indigenous populations in research is imperative, given the established health inequities that Indigenous peoples face worldwide and their current underrepresentation in the literature. Despite many similarities in Indigenous peoples' colonization experiences, a nuanced consideration is needed of the various factors that have potential to influence the microbiomes of Indigenous populations.

## THE INTERSECTION BETWEEN INDIGENEITY AND FACTORS THAT IMPACT THE GUT MICROBIOME

### Urbanization

Urban and rural Indigenous peoples have been described as "two facets of the same people" (72). In the modern day, cities are one of the main sites of residence for Indigenous populations. Migration is not a new concept to many Indigenous peoples, whether migrating between countries, within countries, or experiencing changes to the environment itself due to events like climate change and urbanization. Increasing migration from rural areas is being seen across the globe due to factors including loss of land, climate change, poverty, and the pursuit of new opportunities (72). Urbanization has been associated with loss of Indigenous culture and identity, particularly due to a lack of attention and effort to meet the needs of urban Indigenous populations, further contributing to disparities (73, 74). It is important that rural, semi-rural, and urban indigenous populations are studied and acknowledged as the differing environmental factors may have implications for data interpretation. When considering the influence of locality, it is necessary also to take into account how bidirectionality (where people move backward and forward between urban and non-urban contexts over the course of their lives) and multi-locality (a lifestyle involving living in several places) may influence the gut microbiome. Currently, remote Indigenous populations have been the subject of more microbiome-focused research compared with those living in cities, such as urban Māori (28, 29, 32–36). Given the scarcity of such research, we acknowledge our reliance here on the use of studies primarily focused on the gut microbiomes of rural and remote Indigenous peoples. Despite more Māori living in urban environments, on average, the prevalence of Māori in rural settings still surpasses that of non-Māori, making representation of rural data important (75). Furthermore, this may offer insights into how colonization might have shifted the Māori gut microbiome over the last 200 years. It is difficult to hypothesize whether "ancestral" species persist in the Māori gut or whether rapid colonization has erased distinctions between Māori and non-Māori gut microbiomes. Over time, it is likely that "ancestral" species in the Māori gut may have become less common or disappeared, as colonization has led to significant changes in urbanization, diet, and health. Depending on the extent of these changes, it is possible that some Māori microbiomes may have characteristics of both pre-colonial and western microbiomes. In turn, this could inform how colonization has impacted health over time.

Rural homelands and urbanized cities are contrasting environments, and migration into urban areas can lead to many changes that have the potential to impact the gut microbiome. Rural and remote Indigenous people tend to exhibit higher gut microbiome diversity and compositional differences compared with non-Indigenous living in urban areas (32–36, 76). Although the idea that rural and urban gut microbiomes differ is widely accepted, it is less clear why this is so, particularly in Indigenous populations. Continuing to work with Indigenous people, like Māori in NZ, to understand their lived experiences in these environments may break down any misconceptions and has the potential to generate learnings that may be unexpected because Indigenous perspectives were centered. Urbanization leads to changes in diet, SES, and health (27–29, 32, 34, 35, 77). As these factors directly link to changes in the gut microbiome, urbanization likely indirectly impacts the gut microbiome (Fig. 1).

In general, urbanization encourages a more Western diet, which contributes significantly to changes in gut microbiome composition and diversity (29, 35, 78–80).

This has been shown eloquently in a Mexican study where rural living Indigenous Me'phaa children were compared with non-Indigenous Mexican children from Mexico City (35). The gut microbiome of the Indigenous children showed significant differences with greater microbial diversity and higher rates of VANISH (volatile and/or associated negatively with industrialized societies of humans) bacterial families compared with the gut microbiomes of the non-Indigenous children who demonstrated a higher relative abundance of BloSSUM (bloom or selected in societies of urbanization/modernization) taxa. These changes reflect the diet of each population, with the Me'phaa diet being high in fiber, consisting of gathered legumes, edible plants, and hunted animal protein (35). In contrast, the westernized Mexican diet is rich in store-bought domesticated animal protein, refined oils, sugars, and limited dietary fiber and vegetables. However, we cannot generalize the findings of urban non-Indigenous populations to urban Indigenous populations due to other factors specific to urban Indigenous peoples. For example, Indigenous food insecurity in urban populations is significantly higher than for those living in rural areas, due at least in part to the erosion of traditional foods from the diet, which may have further microbial impacts (78). Therefore, more literature examining the dietary differences and, subsequently, the gut microbial changes of Indigenous peoples over the course of urbanization is necessary.

Currently, there are limited data pertaining to the combined effect of SES and urbanization on the gut microbiome. Indigenous populations are significantly more likely to live in rural areas than their non-Indigenous counterparts (75, 81, 82). Rural and remote living is often associated with a lower SES (83). A Dutch study revealed associations between income, rural living environments, and the gut microbiome (77). Furthermore, childhood living environment, depending on urbanization level, significantly shaped the adult gut microbiome (77). The extent of rurality is also impacted by Indigenous status. For example, when Māori live in rural areas, they are considerably more likely to live in remote places than non-Māori (75). It is likely that urbanization and SES affect the gut microbiome in Indigenous populations; however, more research is required to understand these associations.

Globally, the health burden of urban and rural peoples differs. Rural residents tend to have less access to healthcare and more limited health literacy compared with their urban counterparts (84). This disparity is likely amplified in Indigenous peoples living rurally, as the mortality of rural Indigenous peoples is often higher (75, 83). To examine how urbanization affects the health and gut microbiome of the Orang Asli, the oldest Indigenous population from Peninsula Malaysia, three different populations living in different degrees of urbanization were studied: urban, semi-urban (rural), and semi-nomadic (remote) hunter gatherers (32, 34). Urbanized Orang Asli had poorer cardiometabolic health measurements compared with the other groups, with each group having distinct gut microbiota profiles. The semi-urban Orang Asli microbiome profiles shared properties of both semi-nomadic and urban gut microbiomes, perhaps representing a compositional shift in the gut microbiome over the course of urbanization. These studies provide insights into how health and the gut microbiome are impacted by urbanization. Interestingly, more uncharacterized microbial genomes were found in the semi-nomadic Orang Asli populations than in the urban populations (32, 34). Thus, these observations point to the increased diversity and lack of characterization of species found in remote societies. This acts as another example of intrapopulation heterogeneity dependent on geographic location and lifestyle.

Urbanization is likely to impact gut microbiomes of Indigenous peoples differently, due to the many disparities they face that heighten or reduce over the course of urbanization. This relationship is extremely complex, as although rural living pre- or early colonization may have posed benefits, rural living now is associated with negative health outcomes. For example, ready access to healthcare services in NZ is increasingly fraught for those living outside major centers, such that Māori are essentially suffering where once they thrived. The scale has tipped in such a way that there is no winning for Māori, as health outcomes have worsened due to urbanization, but now rural living

in NZ is no longer a better outcome due to colonization and the disparities faced by today's rural dwellers. Although it is clear that diet, SES, and the health of Indigenous peoples affect the gut microbiome, it is important to establish that urbanization further influences these factors, indirectly impacting the microbiome (Fig. 1). The following sections will examine these factors and their direct link to gut microbial ecology in Indigenous peoples.

## Diet

The social functions and aspects of food are especially significant for Indigenous peoples, in addition to the universal central value of food as a necessity of life. Food and diet unite individuals through the harvesting, preparation, and consumption of food, which can support many Indigenous cultures in feeling connected to nature and the land (85, 86). Food also facilitates the gathering of families, which promotes the transmission of skills across generations as well as the maintenance of spirituality (86). Within Māori culture, for example, the values of whanaungatanga (sense of family connection) and manaakitanga (kindness and hospitality) are incorporated into sharing kai (food) (85). The dietary patterns of different ethnicities and cultures can vary significantly, particularly the traditional diets of Indigenous cultures versus the typical industrialized diet (87, 88). In addition to food contributing to cultural preservation, diet plays a large role in affecting microbial diversity and composition within the gut, and examining this link in underrepresented populations can broaden our knowledge of microbes associated with people (89, 90).

It is likely that the Māori gut microbiome has undergone vast changes since colonization by Europeans just under 200 years ago. The archetypal pre-colonization Māori diet is thought to have been nutrient-rich and well-balanced (87). This diet included foods high in resistant starch like kūmara and taro, native leafy greens, berries, and seeds. Colonization has allowed access to foods representative of a highly-processed Western dietary pattern, which is high in fat, sodium, and sugar, but low in nutrient-rich foods like fruits, vegetables, and lean proteins (91). Furthermore, sanitation of Western foods has likely influenced the modern Māori gut microbiome, as meats and vegetables no longer act as such influential environmental reservoirs for microbes to colonize the gut, which still plays a role in shaping the gut microbiome of some Indigenous hunter-gatherers (28). Nonetheless, the modern diet of the Māori population differs from the diet of non-Māori within NZ, particularly in total consumed macronutrients (92). Māori men tend to eat more protein, cholesterol, and mono-unsaturated fats than non-Māori NZ men (92). Compared with non-Māori women, Māori women are reported to eat more protein, carbohydrates, starch, saturated fatty acids, and polyunsaturated fats. Furthermore, overall daily Māori energy intake is higher (92). Such macronutrients can alter gut microbiota composition, as they can act as substrates for microbial species (90). In addition, Māori tend to eat less vegetables than non-Māori, which may also impact the gut microbiome (92, 93). This variance in diet and macronutrient intake could lead to a significant difference between Māori and non-Māori NZ gut microbiomes.

To date, no studies have examined the relationship between Māori gut microbiomes and diet; however, a limited number of studies suggest that diet impacts the gut microbiomes of Indigenous populations (27, 28, 30, 37). One such study examined the association between diet, metabolic health, and the gut microbiomes of Torres Strait Islander populations from two distinct geographic locations, Waiben and Mer (27). Mer people reside further from the Australian mainland, limiting access to Western foods and resulting in a more traditional, seafood-rich diet. By contrast, Waiben, a more accessible island, allows a flow-through of Western food including takeaways and alcohol. Alcohol consumption, in particular, impacts the gut microbiome and is often a confounder in gut microbiome studies (94). The Torres Strait Islander study reported differences in species-level composition and microbial evenness between the islands (27). In the Mer population, species across five bacterial phyla were enriched, including the opportunistic pathogens *Klebsiella pneumoniae* and *Escherichia coli*. Associations between aspects of

the diet (i.e., sugar-sweetened beverages) and inflammatory markers (i.e., interleukin (IL)−15) were also described (27). A later section of this perspective will focus on the associations between diet, gut microbiome, and Indigenous health.

Remote hunter-gatherer Indigenous populations face seasonal dietary and other environmental changes, which impact the gut microbiome (28). In Tanzania, the consumption of foods that comprise the Hadza hunter-gatherer diet, such as honey, berries, fruits, starchy vegetables, and meat (28), is heavily impacted by season; for example, more meat is eaten in dry seasons, whereas honey availability is higher in wet seasons. This leads to dramatic changes in operational taxonomic units (OTUs) in the *Bacteroidetes* phylum between seasons (28). Furthermore, microbes involved in carbohydrate degradation are seasonally volatile. Many such microbes are no longer present in modern industrialized microbiomes, despite being important for nutrient metabolism. Like Hadza, Canadian Inuit are traditionally hunter-gatherer peoples; however, a transition to a more processed Western diet is occurring in these populations due to increasing access to supermarkets (29). Despite this, seasonal hunting trips are still in practice and much of the Inuit diet is composed of fat-rich, hunted meats. Gut microbiomes of individuals within the Inuit population vary considerably in terms of alpha and beta diversity over time. Furthermore, the inter-individual variance was higher than in Westernized individuals from Montréal, perhaps reflecting the individual opportunistic practices of gathering and hunting (29). Both studies provide insights into how the gut microbiomes of different Indigenous ethnicities respond to seasons, based on diet, geography, and access to Western foods.

Based on the findings above, one may speculate that the diet (and therefore the gut microbiome) of Māori hunter-gatherers changed significantly after their migration to NZ to reflect seasonal availability of food and the introduction of new foods by European settlers. Initially, Māori consumed cultivated vegetables like kūmara, yams, and taro, as well as a range of wild foraged plants (e.g., berries, ferns, seeds) and animals (e.g., fish and native birds) (95, 96). Due to contrasts in geographic climate between the warmer north and cooler south of NZ, it is likely that differences in crop availability caused some contrasts in diet among iwi due to the less successful cultivation of certain crops (e.g., kūmara and taro) further south (96). Also of note, the maramataka (Māori lunar calendar), which was adapted upon arrival to accommodate the seasons and climate of NZ, mayhave indirectly impacted gut microbiomes due to its ubiquitous traditional and current use to guide hunting, gathering and crop planting, and harvesting activities (97, 98). European settlers introduced versatile crops like potatoes (which were easier to cultivate than kūmara), maize, flour, and animals like pigs, sheep, and chickens (95). These introduced foods allowed the creation of Māori breads like rēwena (sourdough) and takakau (flatbread), new fermented foods (e.g., kānga wai; fermented corn), and the consumption of new protein sources (95, 97). The combination of seasonal cultivation, climate differences based on geography, and large dietary pattern changes likely caused major shifts in gut microbiome composition of Māori. However, it can be hypothesized that over the continued course of colonization, transition to western practices and diet further shifted the gut microbiome and has likely diminished seasonal impacts, especially for Māori who do not have access to their traditional dietary patterns and live in urban centers.

Measures of dietary quality in Indigenous peoples also provide insights into the impact of diet on the gut microbiome. An observational cohort study on the gut microbiomes of Native Hawaiians and Pacific Islanders demonstrated that dietary quality impacted specific taxonomic changes in the gut microbiome (30). Furthermore, abundance of the short-chain fatty acid (SCFA) butyrate synthesis gene (butyrate kinase) was positively and negatively associated with multiple bacterial genera and bacterial butyrate kinase gene abundance and expression (30). This finding indicates that the gut microbiome is also important in the microbial production of SCFAs in response to diet in Native Hawaiians and Pacific Islanders (99). Positive and negative associations were also established between dietary quality and SES; as Indigenous populations often suffer

poverty and low SES, making healthy and nutritious food choices can be challenging due to the cost of food (85, 100). Another study analyzing the effects of obesity on gut microbiomes of Pacific Island and NZ European women revealed that distinctive microbiome enterotypes are reflective of dietary intake between both populations (37). The enterotype more strongly associated with Pacific Island women was characterized by higher relative abundances of lactic acid-forming bacterial species, with these women more likely to consume sugar sweetened beverages and discretionary savory foods (e.g desserts, processed meats, and confectionary) (37). The enterotype predominantly associated with NZ European women was characterized by higher relative abundances of *Subdoligranulum* sp., *Akkermansia muciniphila*, *Ruminococcus bromii*, and *Methanobacter smithii* and these women tended to have higher dietary intakes of non-starchy vegetables, nuts and seeds, and plant-based fats (37). This further highlights the discrepancy between Pacific Islanders and NZ Europeans in terms of dietary quality, and how it influences the gut microbiome. If the same analyses with Māori and NZ European women were performed, we might expect a similar result due to their general differences in dietary pattern (92). In NZ, recent increases in the food price index (101) are likely to promote the purchase of less expensive, heavily processed foods for many Māori and Pasifika.

As Māori tend to eat differently to other populations in NZ, it is reasonable to assume that this impacts their gut microbiome and could lead to significant taxonomic and functional differences between the gut microbiomes of Māori and non-Māori. Moreover, it is likely that the Māori gut microbiome underwent drastic changes over the last two centuries due to rapid urbanization altering Māori eating habits, food gathering styles and sanitation over a short time frame. This phenomenon has been seen in a limited number of observational studies in other Indigenous populations (27–30). However, interventional studies are needed to examine whether diet impacts the gut microbiome of Indigenous peoples differently from other populations. To date, observational studies often extrapolate from one population to another rather than tracking dietary differences over a period of time, which would assess how particular diets impact the gut microbiomes of Indigenous peoples.

## Socioeconomic status

Indigenous populations across the globe experience higher rates of lifestyle and income-related disparities and have a lower human development index compared to their non-Indigenous counterparts (15). This is despite the countries in question often ranking among the top 10 globally in terms of human development index (15). Indeed, migration to a wealthier country can, somewhat counter-intuitively, be associated with an altered gut microbiome linked to worsened health for immigrants (102, 103). Using census data, a large cohort study measured multiple disparities across Indigenous populations in NZ, Australia, and Canada (104). Between 1981 and 2006, overall social and health outcomes were worse for Indigenous compared to non-Indigenous populations in all three countries (104). Specifically, there was a lack of progress to improve education and SES for Indigenous people. Every year, the annual income of Indigenous populations across these countries was less than that of non-Indigenous people. In the NZ population, the employment and education gap between Māori and non-Māori had even increased, with Māori continuing to have a higher rate of unemployment and lower median weekly income compared to non-Māori (105). Compared to NZ Europeans, Māori are also approximately four times more likely to experience household crowding (106), which occurs when the number of inhabitants in a home exceeds the dwelling space available and is a major consequence of poverty and low SES (107). However, it is important to note that household crowding is not always a negative experience. For example, on average, Pacific peoples living in crowded households report lower rates of loneliness, and a greater sense of family connection, which may in turn improve mental health (108). Factors that contribute to SES, such as dietary

differences, health status, and social aspects like household crowding are also thought to impact the gut microbiome (109, 110).

Few studies have explored the relationship between SES and the gut microbiome (111, 112), and only one focusses on an Indigenous population (31). This Israeli study examined the gut microbiomes of Indigenous, Arab children across three villages of differing SES (31). Household SES was significantly associated with the gut microbiomes of children, with bacterial diversity being higher (but with reduced evenness) in the village with the lowest average household SES and there being distinct compositional differences between the low SES village and the two higher SES villages (31). A strong positive association between household crowding and species richness was established, as well as significant compositional differences based on household crowding index. This is an interesting paradox as, while household crowding is often seen as a detriment to health, an increase in gut microbial richness may have positive health outcomes (113). The authors hypothesize that potential reasons for these differences include the "old friends" hypothesis, where having a larger family may benefit the immune system through increased microbial exposure impacting the gut microbiome (31, 114). Alternatively, the increased bacterial species may reflect longer contact with hosts for colonization, which is enhanced by household crowding.

A factor that is strongly associated with SES is education, as schools in low socioeconomic communities often lack proper support and funding, and parents may not be able to afford an adequate education for their children (115). Upon adjusting for gender, age and BMI z-score, the Koala Birth Cohort Study, comprised of 281 Dutch children, suggested that maternal education level was associated with changes in the gut microbiome composition among their children, pointing to a potential role for education level in influencing the gut microbiome profile (111).

The links between SES and the gut microbiome are consistent across populations, with data from the Guangdong Gut Microbiome Project showing that up to 40% of detected OTUs were associated with economic status and spending amount (112). The main OTUs positively associated with economic status were taxa from the genus *Bacteroides*, while the relative abundance of the genus *Prevotella* was negatively associated with SES. This is an interesting finding as *Prevotella* species are often enriched in Indigenous populations and negatively associated with westernization (34, 35).

SES impacts other gut microbiome-modifying factors, such as breastfeeding, which is less actively initiated and occurs in shorter durations on average in mothers with a lower SES (116, 117). Breastfeeding impacts the gut microbiome profiles of infants, and potentially even influences those of adults later in life (118, 119). Typically, Māori and other Indigenous peoples tend to breastfeed their babies less than non-Indigenous people, which is particularly significant in urban environments (120–122). Breastfeeding likelihood is impacted by SES and household incomes (121, 123). Reasons for this include misinformation surrounding the benefits of breastfeeding versus bottle-feeding, being shamed for breastfeeding in public, or the stress of living in a low-income household leading to working mothers having less time for breastfeeding (124, 125). However, there is currently no literature on the combined effect of SES and breastfeeding on the gut microbiome, and certainly not in Indigenous peoples. There is no one reason as to why SES impacts the gut microbiome, but likely multiple correlated environmental factors are involved (126). It is reasonable to assume these factors are probably different for Indigenous compared to non-Indigenous populations due to differences in socioeconomic and other disparities.

There is evidence that SES could impact the gut microbiome of Indigenous peoples (31). The SES effect on Indigenous microbiomes is particularly understudied, despite Indigenous populations being especially impacted by SES disparities. Therefore, as Māori are more likely to reside in a low household SES and be exposed to household crowding, this may cause changes to the gut microbiome of Māori compared to non-Māori, as has been previously demonstrated (31).

## Health and wellness

Of all the disparities faced by Indigenous populations, those related to health are the most well-studied. Indigenous peoples' health is consistently poorer than that of non-Indigenous populations. This is due to numerous social determinants, such as a higher prevalence of poverty and less access to healthcare, especially to culturally safe healthcare environments (45, 127, 128). In Indigenous populations who have faced colonization, these health disparities have evolved over time. With the advent of colonization in NZ in the early 1800s, Māori were introduced to new communicable diseases from Europe, such as measles and influenza, with devastating consequences. As there is evidence that communicable disease can impact the gut microbiome, it is reasonable to hypothesize that introduced disease could have initiated shifts in gut microbiome composition during the initial colonization of NZ (129–131). This may have been through infection of pathogens causing direct changes to gut composition, or a more general deterioration in health leading to indirect changes. Since then, lifestyle changes associated with colonization have contributed to the rise of numerous non-communicable diseases that are associated with major health disparities (20, 21). Even when Māori can access healthcare services, they commonly report being exposed to racism, mispronunciation of their names by non-Māori doctors, difficulty building relationships with doctors, and lack of Rongoā (Māori medical practices) or acknowledgement of its benefits for Māori (45). This leads to poor engagement and less effective healthcare for Māori in general. Māori have a disproportionately higher incidence of cardiovascular disease, Type 2 Diabetes Mellitus, metabolic syndrome, and obesity (21, 63, 132). Such non-communicable diseases can lead to changes in the gut microbiome. T2DM has been linked to an altered gut microbiome profile (8, 9), for example differences in the *Firmicutes/Bacteroidetes* phyla ratio have been associated with both T2DM and obesity (133), although this pattern is by no means universal (134). Specific microbial species play a potential protective role against T2DM, such as *Akkermansia muciniphila* (135, 136). Cardiovascular disease may also be linked to the gut microbiome, as microbes can produce metabolites associated with an increased risk of cardiovascular disease (10). With Māori being more likely to develop such diseases, Māori gut microbiomes may differ significantly from those of non-Māori.

Studies indicating that health status leads to changes in Indigenous gut microbiomes are sparse but do make important inferences. Correlations between age, BMI, glycated hemoglobin (HbA1c) and gut microbial richness have been examined in Native Hawaiians and Pacific Islanders (30). Examination of microbial profiles revealed differences in numerous phyla across age groups, notably: *Bacteroidetes, Firmicutes, Actinobacteria, Deferribacteres* and *Proteobacteria*. Certain species within the *Bacteroidetes* phylum also displayed a negative correlation with glycemia and obesity, while the genus *Bifidobacterium* was negatively correlated with age, HbA1c, and BMI, pointing to its positive associations with health in Native Hawaiians and Pacific Islanders (30). Unlike in other populations where T2DM is most closely associated with BMI, in this cohort age had a greater positive association with T2DM than BMI. As this finding contradicted previous research undertaken in other ethnicities, the importance of studying Indigenous populations to highlight potential ethnic or culture-specific relationships is evident. This observation supports the contention that findings in majority, westernized ethnicities may not be generalizable to minority, Indigenous populations, and emphasizes the importance of Indigenous-focused studies.

As noted previously, differences between the gut microbiomes of Torres Strait Islanders from Mer and Waiben highlighted how particular bacterial genera or species may shape health (27). Interestingly, despite having a "less-western" diet, Torres Strait Islanders residing in Mer had a higher mean arterial blood pressure with specific bacteria being associated with blood pressure. For example, *Lachnospiraceae* bacterium 8_1_57FAA mediated a positive relationship with mean arterial blood pressure and interleukin-15 (IL-15) concentrations. IL-15 is positively associated with cardiovascular disease (137) and gut dysbiosis, and can alter the gut microbiome (138). Furthermore,

*Alistipes onderdonkii* had a relationship with systolic blood pressure (27). In the Mer population, systolic blood pressure was increased in the absence of *A. onderdonkii*. In contrast, in *A. onderdonkii*-positive Waiben residents, there was no difference in systolic blood pressure (27). This shows that associations between specific gut microbes and health or disease may be mediated by geographic location and other lifestyle factors, and not just based on their Indigenous ethnicity, further displaying Indigenous intrapopulation heterogeneity.

Communicable disease in Indigenous populations, particularly disease-associated gut dysbiosis, also impacts the gut microbiome. A study in the Orang Asli demonstrated that eukaryotic helminth infection was associated with increased gut microbiome diversity and increased with the severity of infection (32). Specific bacterial associations with helminth infection were village dependent, further showing how geographic location may impact specific bacterial relationships with health factors (27). A similar study examined members of Wiwa, an Indigenous Colombian tribe, who suffer high instances of gut infection (33). Associations were found between relative abundances of specific bacterial genera and gut infections. For example, enterohemorrhagic *Escherichia coli* infection was positively associated with the abundance of *Sutterella* species, and relative *Firmicutes* abundance was associated with infection by enteropathogenic *E. coli* and helminth species. This research also shows how communicable pathogens such as helminths can impact the gut microbiome in Indigenous populations and how medications may further impact the gut microbiome.

While it is clear that physical health has a significant connection to the gut microbiome, the association between mental health and the gut microbiome remains unclear. Previous research has shown a bidirectional link between the gut and the brain, known as the gut-brain axis (139), with communication occurring through factors of the nervous, endocrine, and immune systems (139). While having no specific link to Indigenous populations, a study examining discrimination between different ethnicities found significant changes in brain resting state connectivity, as measured by functional magnetic resonance imaging, associated with specific species in the gut microbiome (140). Particularly, *Prevotella copri* was strongly positively associated with discrimination in Black and Hispanic individuals. Discrimination was linked with altered brain connectivity and increased inflammation, which can lead to increased reactive oxygen species that *P. copri* can utilize to survive in inflammatory environments (140). This bacterium is associated with worsened health outcomes such as rheumatoid arthritis, displaying the link between mental health, the gut microbiome, and physical health (141). Māori and Indigenous populations have increased rates of mental health disorders and are subject to more racial discrimination (142, 143). The downstream effects of colonization, urbanization and cultural disconnection of Māori are believed to play key roles in these increased mental distress rates (144). Therefore, the specific effects of discrimination and worsened Indigenous mental health on the gut microbiome should be investigated.

Due to a lack of accessible healthcare, other factors such as immunization and antibiotic use are likely to impact the gut microbiomes of Indigenous populations. In NZ, on average, Māori display higher vaccine hesitancy, alongside having less access to healthcare, which has led to decreased vaccine uptake in Māori populations (145). Vaccinations are thought to have a bidirectional relationship with the gut microbiome (146). Specifically, COVID-19 vaccines can decrease gut microbiome diversity, confer compositional shifts and impact specific microbial species, and baseline gut microbiome composition can contribute to adverse effects of the vaccines (147). Other xenobiotics such as antibiotics can also impact the gut microbiome, especially those which are broad spectrum (148). In NZ, inappropriate prescription of antibiotics is particularly prevalent compared to other nations (149). Māori are faced with both the potential of being over-prescribed and under-prescribed antibiotics: on the one hand, rates of prescriptions to Māori are higher than for non-Māori populations, however disease burden in Māori is also disproportionately higher (150). This implies a potential inclination to over or under prescribe antibiotics in specific cases, necessitating further investigation to ensure Māori

are receiving proper treatment. In either case, this could impact the gut microbiome of Māori in NZ. More research on antibiotic usage and Māori gut microbiomes is required to truly understand this interplay.

In general, Indigenous peoples suffer greater health disparities than their non-Indigenous counterparts. These disparities can influence gut microbiome diversity and composition, which may contribute to worsened health outcomes in these populations. While it is well documented that Māori face health disparities, literature on the connection between Māori, the gut microbiome and health has yet to be published.

## CONCLUSION

Globally, human gut microbiome research conducted by and for Indigenous populations is currently lacking. Utilizing Indigenous science, like Mātauranga and Kaupapa Māori methodologies, may benefit the microbial ecology space through the formation of new knowledge, and start to break down social and cultural barriers to Indigenous participation in human clinical trials. Indigenous peoples tend to have differing diets and, on average, suffer greater SES and health disparities compared to non-Indigenous peoples, primarily due to the downstream effects of colonization and urbanization. Such disparities may play an influential role in shaping the gut microbiomes of Indigenous peoples, both urbanized and non-urbanized. As gut microbiome research with Indigenous populations is becoming more common, the implementation of robust and culturally-responsible research ethics, in parallel, is becoming increasingly important. There remains a significant need to deconstruct and further understand the impact of disparities on the gut microbiome and how associated microbial changes impact the health of Indigenous peoples. While it is apparent that we cannot generalize findings between Indigenous and non-Indigenous peoples, it is also true that Indigenous populations are heterogenous, and care should be taken when comparing Indigenous groups to one another. Research led by and involving specific Indigenous populations, like Māori, is required to elucidate these differences, and the implications this may have on the gut microbiome.

## ACKNOWLEDGMENTS

We would like to acknowledge the efforts of our reviewers in improving this paper by providing detailed suggestions and perspectives on this important and relevant topic. It is clear that all of you have great sensitivity and passion within this field of work, and you have contributed extensively to this perspective piece.

## AUTHOR AFFILIATIONS

[1]Department of Medicine, University of Otago, Christchurch, New Zealand

[2]High-Value Nutrition National Science Challenge, Auckland, New Zealand

[3]Edible Research, Ohoka, New Zealand

[4]Department of Human Nutrition, University of Otago, Dunedin, New Zealand

[5]Riddet Institute, Palmerston North, New Zealand

[6]School of Biological Sciences, University of Auckland, Auckland, New Zealand

[7]Institute of Biotechnology, Helsinki Institute of Life Science (HiLIFE), University of Helsinki, Helsinki, Finland

[8]Department of Microbiology, Faculty of Agriculture and Forestry, University of Helsinki, Helsinki, Finland

[9]Research Program for Clinical and Molecular Metabolism, Faculty of Medicine, University of Helsinki, Helsinki, Finland

[10]Liggins Institute, University of Auckland, Auckland, New Zealand

## AUTHOR ORCIDs

Ella T. Silk  http://orcid.org/0009-0000-2598-9220

Richard B. Gearry  http://orcid.org/0000-0002-2298-5141

## FUNDING

| Funder | Grant(s) | Author(s) |
|---|---|---|
| University of Otago (Te Whare Wānanga o Otāgo) | Doctoral Scholarship | Ella T. Silk |
| Manatu Hauora \| Health Research Council of New Zealand (HRC) | HRC 23/314 | Ella T. Silk |
| | | Nicole C. Roy |
| | | Meika Foster |
| | | Michael W. Taylor |
| High Value Nutrition (New Zealand) | UOAX1421,UOAX1902 | Ella T. Silk |
| | | Simone B. Bayer |
| | | Nicole C. Roy |
| | | Meika Foster |
| | | Michael W. Taylor |

## AUTHOR CONTRIBUTIONS

Ella T. Silk, Conceptualization, Writing – original draft, Writing – review and editing | Simone B. Bayer, Supervision, Writing – review and editing | Meika Foster, Writing – review and editing | Nicole C. Roy, Funding acquisition, Project administration, Supervision, Writing – review and editing | Michael W. Taylor, Supervision, Writing – review and editing | Tommi Vatanen, Writing – review and editing | Richard B. Gearry, Supervision, Writing – review and editing

## ADDITIONAL FILES

The following material is available online.

### Open Peer Review

**PEER REVIEW HISTORY (review-history.pdf).** An accounting of the reviewer comments and feedback.

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
