## [Reviewer comments · mSystems]

Advancing microbiome research in Māori populations: Insights from recent literature exploring the gut microbiomes of underrepresented and Indigenous peoples

Ella Silk, Simone Bayer, Nicole Roy, Meika Foster, Michael Taylor, Tommi Vatanen, and Richard Geary

Corresponding Author(s): Richard Geary, University of Otago Christchurch

Review Timeline:

Submission Date:	July 8, 2024
Editorial Decision:	August 7, 2024
Revision Received:	September 10, 2024
Accepted:	September 11, 2024

Editor: Suzanne Ishaq

Reviewer(s): Disclosure of reviewer identity is with reference to reviewer comments included in decision letter(s). The following individuals involved in review of your submission have agreed to reveal their identity: Aviâja Lyberth Hauptmann (Reviewer #1); Kieran O'Doherty (Reviewer #3)

Transaction Report:

DOI: <https://doi.org/10.1128/msystems.00909-24>

Re: mSystems00909-24 (Diet and Disparities in Shaping Indigenous Microbiomes: The Unexplored Māori Gut Microbiome)

Dear Prof. Richard B Geary:

Revision Guidelines

Sincerely,
Suzanne Ishaq
Editor
mSystems

Reviewer #1 (Comments for the Author):

I would like to commend the authors for taking seriously my encouragement to put research practise and ethics at the center of the manuscript.

In alignment with this I recommend the authors consider revising the title to match the central theme of the paper as also expressed in the abstract, focusing more on microbiome research and Indigenous peoples more broadly rather than diet and

disparities.

Following are specific comments for the manuscript:

Line 22: Throughout the manuscript I suggest not using "Indigenous microbiome research" as it is unclear what is meant - research by or on Indigenous peoples or both?. I suggest being clear about which is meant by stating specifically "research on Indigenous peoples' microbiomes" or "research by Indigenous peoples".

Line 23-24: I suggest making more clear how disparities are central to the theme. Perhaps rephrasing to "... face in NZ, which are likely to lead to gut microbiome differences ..."

Line 25: I suggest "underrepresented populations has the potential to and should lessen health inequities"

Line 26: In line with comment for line 22 I suggest using "the microbiomes of Indigenous peoples" rather than "Indigenous microbiomes" and suggest that the sentence be altered to:
"...the microbiomes of Indigenous peoples, study design must be improved to protect the rights and interests of Indigenous peoples, which will ensure ethical conduct of research and increase participation and beneficial outcomes."

Something to consider for the Introduction section generally: the introduction is quite broad and repeats well-established facts. The first two paragraphs (Lines 33-46) could be made stronger by reducing the general facts and instead emphasizing themes with specific relevance to the point of the Perspective, namely the remarkable importance of the human microbiome for health in its broadest sense including physical and mental, listing the diverse diseases linked to the microbiome as well as pointing out the diverse and context/host specific nature of the microbiome, this way highlighting the importance of understanding this crucial aspect of health and wellbeing in specific context.

Line 33: Consider leaving out "Commensal"

Lines 47-56: I encourage the authors to revise this section to think beyond ethnicity/genetics to include culture, history, questions of relevance, context etc. Indigenous peoples represent different microbe-human relations underrepresented in science today that we can all learn from, not just because of ethnicity.
In line with this, do not use "research ON Indigenous populations" but normalize using "with" instead.

Lines 62-65: This section could use some introduction. I suggest to add, before this section, an introduction saying that Māori have a long history with scientific research and have been leading efforts to decolonize research and call for better scientific research with, by and for Indigenous peoples (Tuhawai Smith 1999).

Line 66: use "the gut microbiomes of Indigenous peoples"

Line 74: rather than "ethnicities" I suggest writing "contexts" as it is much more complex than ethnicity alone.

Line 80: I appreciate the authors including a positionality but would encourage to include one for all authors. I also very much appreciate that this Perspective is written with Indigenous leadership. I do encourage that we do not suggest that as Indigenous peoples we can uncritically represent our communities' world views (see as example Tuck and Yang, 2014, R-Words: Refusing Research page 242, last paragraph). This type of work requires a lot of humility and I suggest not stating that one person can represent the world views of a community.

Line 124-125: Participant numbers and data types are not the only limitations, important limitations are whether the studies are conducted to answer relevant questions respectfully and whether they have the potential to teach us something we didn't expect because Indigenous perspectives were centered.

Lines 133-136: We must refrain from thinking of this as a dichotomy, as this does not represent the truth well. Many people who live far from urban and academic centers eat diets that are far from their ancestors' diets, while some who live in cities with higher incomes can afford more ancestral diets. Also, "remote" is contextual. What is remote to an academic center is central to someone else.

Line 137: The section from here is quite long and has a lot of important points. Perhaps it would improve readability to make a bit more concrete and insert a roadmap of what the next sections are presenting.

Pages 8-11 are excellent and so important. Remember that gut microbes do not just respond to the food eaten, but the food eaten might also bring microbes.
I suggest including a table that lists the important Māori concepts for the reader to be able to return to as they are used in the text, if appropriate to do so.

Line 183: Concepts that align well with an understanding of the holobiont, which has only recently gained traction in the scientific

community.

Section on Urbanization: remember again that this is not unidirectional or a dichotomy. Many people move back and forward between urban and non-urban contexts.

Line 300: I suggest using ", making representation of rural data important (69)."

Line 304-305: I'm having trouble understanding this sentence in this context.

Line 309: consider exchanging "harbor" with "has"

Line 312 "it is less clear why": perhaps lived Māori experiences can help to break down the dichotomous misunderstanding and add important nuances and understanding.

Section starting at line 360-376: this seems not to fit in well at this point in the manuscript. Consider moving.

Line 400: an opportunity to broaden the diversity of microbes known to be associated with people.

Line 450: This section would benefit from more specific information on Māori diet seasonality. The Diet section is not as strong as the following SES section from page 20. It could be more specific.

The section on Health and Wellness is missing a brief discussion of how diseases brought during initial colonization might have impacted the microbiome of Māori. Other than that it reads really well.

Line 642-643: I suggest emphasizing research ethics here, as it is increasingly important as the research in increasing.

Reviewer #3 (Comments for the Author):

Thank you for inviting me to review this revised manuscript. I feel that the authors have addressed my concerns and I congratulate them on this important work. I have only one minor suggestion:

The authors state that: "in other words, due to effects that are class-based rather than race-based per se. While the effects of class and race often go hand-in-hand, there is debate as to which has the biggest impact on Māori in poverty (34), both of which change influential environmental factors like SES (further impacting living conditions), diet, and health, and therefore gut microbial ecology" (p. 5. L 108-112).

I understand the point about class and I think this is an important issue to raise. However, I urge the authors to remove or change their reference to race. Internationally, there has been strong opposition to scientific racism from many directions. There is widespread agreement that race is not a meaningful biological category for humans, and that it was developed historically to oppress people. Of course, the effects of racialization and racism are with us today, and it is important to study and resist this. A part of this resistance can come with avoiding use of the term "race" where it can be interpreted by some as a biological category that serves to explain "racial differences", and instead to highlight the social and institutional circumstances that lead to negative outcomes for racialized groups. To avoid giving a rhetorical opening to those who would use it to further scientific racist end, I suggest using different language that highlights the social foundations of differential health (and other) outcomes across "races". For example, the text above could be changed to "...due to effects that are class-based rather than outcomes of racism. While the effects of class and racism often go hand-in-hand..." or similar.

Response to reviewer comments:

Reviewer #1 (Comments for the Author):

I would like to commend the authors for taking seriously my encouragement to put research practise and ethics at the center of the manuscript.

In alignment with this I recommend the authors consider revising the title to match the central theme of the paper as also expressed in the abstract, focusing more on microbiome research and Indigenous peoples more broadly rather than diet and disparities.

Once again, we thank this reviewer for taking the time to thoroughly review and improve our work, and for believing in the importance of this topic. We think the final changes made have further improved the manuscript. Our replies to each point are bolded.

Following are specific comments for the manuscript:

1. Line 22: Throughout the manuscript I suggest not using "Indigenous microbiome research" as it is unclear what is meant - research by or on Indigenous peoples or both?. I suggest being clear about which is meant by stating specifically "research on Indigenous peoples' microbiomes" or "research by Indigenous peoples".

Agree, we have adjusted this in the abstract and throughout the manuscript.

2. Line 23-24: I suggest making more clear how disparities are central to the theme. Perhaps rephrasing to "... face in NZ, which are likely to lead to gut microbiome differences ..."

Line 25: I suggest "underrepresented populations has the potential to and should lessen health inequities"

Line 26: In line with comment for line 22 I suggest using "the microbiomes of Indigenous peoples" rather than "Indigenous microbiomes" and suggest that the sentence be altered to: "...the microbiomes of Indigenous peoples, study design must be improved to protect the rights and interests of Indigenous peoples, which will ensure ethical conduct of research and increase participation and beneficial outcomes."

We agree with the three points above, in effort to stay within the word limit we have adjusted the abstract to present these same ideas, but in different wording (lines 23 – 29).

3. Something to consider for the Introduction section generally: the introduction is quite broad and repeats well-established facts. The first two paragraphs (Lines 33-46) could be made stronger by reducing the general facts and instead emphasizing themes with specific relevance to the point of the Perspective, namely the remarkable importance of the human microbiome for health in its broadest sense including physical and mental, listing the diverse diseases linked to the microbiome as well as pointing out the diverse and context/host specific nature of the microbiome, this way highlighting the importance of understanding this crucial aspect of health and wellbeing in specific context.

Agree, we have added a line listing some diseases (line 45 – 50) and have also removed unnecessary sentences.

4. Line 33: Consider leaving out "Commensal"

Agree, removed at line 34.

5. Lines 47-56: I encourage the authors to revise this section to think beyond ethnicity/genetics to include culture, history, questions of relevance, context etc. Indigenous peoples represent different microbe-human relations underrepresented in science today that we can all learn from, not just because of ethnicity.

In line with this, do not use "research ON Indigenous populations" but normalize using "with" instead.

Agree, added (lines 60 – 65).

6. Lines 62-65: This section could use some introduction. I suggest to add, before this section, an introduction saying that Māori have a long history with scientific research and have been leading efforts to decolonize research and call for better scientific research with, by and for Indigenous peoples (Tuhiwai Smith 1999).

Agree, this introductory sentence has been added (line 71 – 75). We thank you for your suggestion.

7. Line 66: use "the gut microbiomes of Indigenous peoples"

Agree (line 78).

8. Line 74: rather than "ethnicities" I suggest writing "contexts" as it is much more complex than ethnicity alone.

Agree (line 87).

9. Line 80: I appreciate the authors including a positionality but would encourage to include one for all authors. I also very much appreciate that this Perspective is written with Indigenous leadership. I do encourage that we do not suggest that as Indigenous peoples we can uncritically represent our communities' world views (see as example Tuck and Yang, 2014, R-Words: Refusing Research page 242, last paragraph). This type of work requires a lot of humility and I suggest not stating that one person can represent the world views of a community.

We definitely agree with this, not all authors have whakapapa, but respect and understand the idea that more research with underrepresented, Indigenous, populations is central to improving outcomes in their health. We have adjusted the text at lines 91 – 97 to clarify this.

10. Line 124-125: Participant numbers and data types are not the only limitations, important limitations are whether the studies are conducted to answer relevant questions respectfully and whether they have the potential to teach us something we didn't expect because Indigenous perspectives were centered.

Agree, we have weaved this in, in a couple of places (line 144 and 342 – 345).

11. Lines 133-136: We must refrain from thinking of this as a dichotomy, as this does not represent the truth well. Many people who live far from urban and academic centers eat diets that are far from their ancestors' diets, while some who live in cities with higher incomes can afford more ancestral diets. Also, "remote" is contextual. What is remote to an academic center is central to someone else.

Agree, we address this at lines 155 – 159.

12. Line 137: The section from here is quite long and has a lot of important points. Perhaps it would improve readability to make a bit more concrete and insert a roadmap of what the next sections are presenting.

Agree, we have edited a paragraph at the beginning of the “bolstering research on gut microbiomes of Indigenous peoples” to roadmap the following sections (lines 134 – 141).

13. Pages 8-11 are excellent and so important. Remember that gut microbes do not just respond to the food eaten, but the food eaten might also bring microbes.

Thank you, we have now mentioned this (line 81, 422 – 425).

14. I suggest including a table that lists the important Māori concepts for the reader to be able to return to as they are used in the text, if appropriate to do so.

Agree (line 167).

15. Line 183: Concepts that align well with an understanding of the holobiont, which has only recently gained traction in the scientific community.

Thank you, we have now mentioned the holobiont in line 211, for completeness.

16. Section on Urbanization: remember again that this is not unidirectional or a dichotomy. Many people move back and forward between urban and non-urban contexts.

Agree (lines 320 – 323).

17. Line 300: I suggest using ", making representation of rural data important (69)."

Agree (line 327 – 328).

18. Line 304-305: I'm having trouble understanding this sentence in this context.

Agree, we have adjusted this for clarity (line 332 - 336).

19. Line 309: consider exchanging "harbor" with "has"

Agree (line 343).

20. Line 312 "it is less clear why": perhaps lived Māori experiences can help to break down the dichotomous misunderstanding and add important nuances and understanding.

Agree (lines 342 – 345).

21. Section starting at line 360-376: this seems not to fit in well at this point in the manuscript. Consider moving.

We have now moved this paragraph back to where it started (lines 652 – 668). A previous reviewer had suggested it be moved into the urbanization section, but we agree that it fits more in the health and wellness section.

22. Line 400: an opportunity to broaden the diversity of microbes known to be associated with people.

Agree (line 413 – 416).

23. Line 450: This section would benefit from more specific information on Māori diet seasonality. The Diet section is not as strong as the following SES section from page 20. It could be more specific.

Agree, we have gone into seasonality and geography of Māori and how this could potentially impact the gut microbiome (line 465 – 484).

24. The section on Health and Wellness is missing a brief discussion of how disease brought during initial colonization might have impacted the microbiome of Māori. Other than that it reads really well.

Agree, we have added some information on this (line 593 – 597).

25. Line 642-643: I suggest emphasizing research ethics here, as it is increasingly important as the research in increasing.

Agree, we have now emphasized this (line 698 – 702).

Reviewer #3 (Comments for the Author):

Thank you for inviting me to review this revised manuscript. I feel that the authors have addressed my concerns and I congratulate them on this important work. I have only one minor suggestion:

The authors state that: "in other words, due to effects that are class-based rather than race-based per se. While the effects of class and race often go hand-in-hand, there is debate as to which has the biggest impact on Māori in poverty (34), both of which change influential environmental factors like SES (further impacting living conditions), diet, and health, and therefore gut microbial ecology" (p. 5. L 108-112).

I understand the point about class and I think this is an important issue to raise. However, I urge the authors to remove or change their reference to race. Internationally, there has been strong opposition to scientific racism from many directions. There is widespread agreement that race is not a meaningful biological category for humans, and that it was developed historically to oppress people. Of course, the effects of racialization and racism are with us today, and it is important to study and resist this. A part of this resistance can come with avoiding use of the term "race" where it can be interpreted by some as a biological category that serves to explain "racial differences", and instead to highlight the social and institutional circumstances that lead to negative outcomes for racialized groups. To avoid giving a rhetorical opening to those who would use it to further scientific racist ends, I suggest using different language that highlights the social foundations of differential health (and other) outcomes across "races". For example, the text above could be changed to "...due to effects that are class-based rather than outcomes of racism. While the effects of class and racism often go hand-in-hand..." or similar.

We again thank this reviewer for their final suggestion and for taking the time to read our manuscript again. We believe this is an important suggestion and have since reworded this section to avoid using the word 'race' due to its negative connotations (throughout lines 125 – 129).

Re: mSystems00909-24R1 (Advancing microbiome research in Māori populations: Insights from recent literature exploring the gut microbiomes of underrepresented and Indigenous peoples)

Dear Prof. Richard B Geary:

Your manuscript has been accepted, and I am forwarding it to the ASM production staff for publication. Your paper will first be checked to make sure all elements meet the technical requirements. ASM staff will contact you if anything needs to be revised before copyediting and production can begin. Otherwise, you will be notified when your proofs are ready to be viewed.

Sincerely,
Suzanne Ishaq
Editor
mSystems